# Robust Identification System for Spanish Sign Language Based on Three-Dimensional Frame Information

**DOI:** 10.3390/s23010481

**Published:** 2023-01-02

**Authors:** Jesús Galván-Ruiz, Carlos M. Travieso-González, Alejandro Pinan-Roescher, Jesús B. Alonso-Hernández

**Affiliations:** 1IDeTIC, Universidad de Las Palmas de G.C. (ULPGC), 35017 Las Palmas de Gran Canaria, Spain; 2Signals and Communications Department, Universidad de Las Palmas de G.C. (ULPGC), 35017 Las Palmas de Gran Canaria, Spain

**Keywords:** dynamic time warping, Spanish sign language, gesture recognition, pattern recognition

## Abstract

Nowadays, according to the World Health Organization (WHO), of the world’s population suffers from a hearing disorder that makes oral communication with other people challenging. At the same time, in an era of technological evolution and digitization, designing tools that could help these people to communicate daily is the base of much scientific research such as that discussed herein. This article describes one of the techniques designed to transcribe Spanish Sign Language (SSL). A Leap Motion volumetric sensor has been used in this research due to its capacity to recognize hand movements in 3 dimensions. In order to carry out this research project, an impaired hearing subject has collaborated in the recording of 176 dynamic words. Finally, for the development of the research, Dynamic Time Warping (DTW) has been used to compare the samples and predict the input with an accuracy of 95.17%.

## 1. Introduction

Based on data retrieved from the United Nations (UN) [1] and considering that a percentage of people with impaired hearing, depending on the area where they live, use at least one of the 300 sign languages that currently exist, there is a need to break down barriers to better integrate hearing-impaired people into a speaking society. Even though most people with a hearing impairment are capable of lip reading, people without a hearing impairment find it challenging to communicate with them. This fact, together with the challenges brought on by the current COVID-19 pandemic, where the use of face masks covering nose and mouth became mandatory in many places, has made it increasingly necessary to develop a system that helps people with this disability. Likewise, the progress in new technologies such as artificial intelligence (AI), sensors, hardware, networks, etc., are improving the lives of many people in different knowledge areas of everyday life. The field of gesture recognition began 70 years ago.

### 1.1. Technologies

In the 1960s, tablets and pens that were capable of capturing writing by using touch-sensitive interfaces or pointing devices came into use. Data gloves, such as the Data Glover [2], also began to be used with the disadvantage that this type of system uses an invasive element. However, these active systems were connected by cables that were a handicap for their versatility. Later, thanks to wireless technologies, these elements were removed, yet the sensors made the glove thicker and, hence, complicated to use.

At the start of the 1980s, other types of systems were developed, such as passive gloves, which had different colors depending on the part of the hand they covered. With these gloves and the vision systems that began to appear at the turn of the decade, together with the best computer systems, hand movements in two dimensions began to be detected.

In the 1990s, there was a very important leap in the creation of new sensors and the improvement of computing equipment. This led to the creation of new active gloves with more precise sensors that were able to reproduce the movement of a hand in a real way. There were also significant advances in the treatment of images and new digital cameras that greatly improved the detection of hand movements in real time [3,4,5]. 

There are also studies which work with electromyography (EMG) and electrodes [6,7,8,9]. These studies are based on the neuromuscular system using electrodes that can detect electrical signals produced by muscles and nerves.

Ultrasound motion-detection systems have been used in several studies, thanks to the Doppler Effect [10,11,12]. Our tissues can endure different acoustic impedances by reflecting different amounts of energy. Nonetheless, this system is not very precise compared to the others mentioned due to losses by occlusion where some parts of the body are left behind others.

The boom of wireless communications began in the year 2000. The system imposed to date, Wi-Fi, has made it possible for the infrastructures and systems in this field to evolve exponentially. Hence, research into gesture recognition was initiated using these networks. However, these systems had the disadvantage of requiring specialized devices or the need of modifying commercial approaches since they were unable to accurately detect hand movements.

Radio frequency identification (RFID) uses commercial devices of Ultra High Frequency (UHF). These systems are mainly used for the automatic management of objects and to control human activities [13,14,15]. RFID also uses phase shifts and the Doppler Effect and is inexpensive. The problem with the RIFD is the detection ranges as they are reduced to a few centimeters.

Currently, the use of RGB cameras is organized in an array of pixels where each one has its value. Based on this capture, the different algorithms are applied for the processing of images. An instance of how they can be applied is by using stereoscopic vision whereby the depth of the view can be calculated by obtaining two different views of the same scene, following the binocular system of the human eye. Active lighting can also be used, and depending on its projection, can establish the depth of the space. This technique is very similar to stereoscopic vision; in fact, it is considered a modification of structured vision.

The use of RGB-d cameras is one of the most common techniques used to obtain depth in video images in real time. These devices are able, through the emitter of infrared light (IR), to obtain the depth of an image. The process that these systems follow is: data acquisition, image pre-processing, segmentation, feature extraction and classification. These cameras began to drop in price when Microsoft’s Kinect came on the market and their use increased [16]. A few years later, Microsoft released the Azure Kinect DK with many more sensors and speech models. This camera was developed for professional use employing AI to detect and study the movements of people and objects.

### 1.2. Relative Works

The progress of electronics and microelectronics has given a very important advance to the field of sensors. This is the case of the Leap Motion [17], a volumetric sensor that is capable of capturing the movement of the hands in three dimensions. There are different projects that use the Leap Motion Controller (LMC) for the recognition of different signs. These can be divided into three trends of use: the use of static signs (dactyl alphabet), the use of static and dynamic signs (words), and finally, only the use of dynamic signs (words).

Regarding the relative works of static signs, Funasaka used 24 static signs that correspond to the alphabet of the American Sign Language [18] (ASL), and the decision tree technique with an accuracy of 82.71%. On the other hand, G. Marin [19] combines LMC and Kinect to recognize 10 ASL signs using a support vector machine (SVM) with an accuracy of 80.86%.

Simos, in [20], used 24 static signs from the Greek Sign Language (GSL), working with multilayer perceptron (MLP), for hands and fingers independently. His accuracy results were 99.08% and 98.96%, respectively. Mapari [21] worked with 32 ASL signs and numbers using MLP with an accuracy of around 90%. Vaitkevičius [22], conversely, worked with Virtual Reality (VR), using Leap Motion 24 ASL static letters in order to create sentences that he later recognized using linear regression analysis with an accuracy of 86.1%.

Among the researchers who developed programs using dynamic and static signs, Mohandes [23,24], for instance, worked with 28 signs of the Arabic Sign Language (ArSL) in a static way, except for two signs that are dynamic. In both experiments, different processing techniques were used, including the K-nearest neighbor algorithm (K-NN), hidden Markov model (HMM), Naïve Bayes and MLP, obtaining an accuracy of 97.1%, 97.7%, 98.3% and 99.1%, respectively. In [25], Hisham worked with 16 ArSL static words with different results depending on the techniques applied, obtaining an accuracy of 90.35% for neural network, 95.22% for K-NN, and between 89.12% and 90.78% for SVM according to the methodology. Hisham also worked with 20 ArSL dynamic words using dynamic time warping, (DTW), with an accuracy of 96.41%. Naglot used 26 ASL letters with MLP and an accuracy of 96.15% [26]. Chong used 26 ASL letters and 10 digits [27]. The results working only with SVM letters were 80.30% and with deep neural network (DNN), 93.81%. The results adding the 10 digits were 72.79% and 88.79% for SVM and DNN, respectively. Lee worked with the 26 ASL letters, two of which are dynamic, using different techniques [28]. For long short-term memory (LSTM), the accuracy was 97.96%, for SVM, 98.35%, and finally for recurrent neural network (RNN), 98.19%. In [29], Tao also used 26 ASL letters with convolutional neural network (CNN), with an accuracy ranging between 80.1% and 99.7%. In [30], Anwar worked with 26 Indonesian Sign Language (SIBI), applying the K-NN and SVM techniques, obtaining a result of 95.15% and 93.85% respectively. Alnahhas [31] worked with 15 ArSL words using LSTM with a result of 96%. Lastly, Avola [32] worked with 18 static signs and 12 dynamic signs using LSTM with an accuracy of 96.41%.

Finally, there are experiments that only used dynamic signs. Among others, in [33], Elons presented an investigation that used 50 words of the Arabic Sign Language ArSL, using MLP, obtaining an accuracy of 88%. On the other hand, Jenkins worked with 13 dynamic words making a comparison with different techniques [34]. With neural network he obtained an accuracy of 99.9%, with random forest, 99.7%, with SVM, 99.9%, K-NN, 98.7% and finally, with Naïve Bayes, a result of 96.4%.

### 1.3. Our Proposal

After studying the state of the art, it can be seen that the use of dynamic signs is the most realistic method to try to identify sign language; however, the number of words used is limited, in the best of cases, to 50 words. In this research, 176 dynamic words from the Spanish Sign Language (SSL) have been used. For the recognition of these words, the DTW has been used, achieving an accuracy of 95.17%. This article presents how the word database has been generated, the methodology used for the experiment, and the results offered by the applied techniques.

An important innovation is to demonstrate that models of signs done by a user can be used by different people while maintaining the level of accuracy. Then, the models can be generated by any person, and these models can be used by any different person. The proposal demonstrates that it is totally independent of the signer.

The objective of the research is to develop a recognition system for Spanish sign language using machine learning techniques. Regarding the research presented in this section, there are some gaps that this article addresses, e.g., the number of dynamic signs, because other related references work with a small dataset vs. this proposal (until 176 dynamic signs).

A second issue that this paper solves is the use of dynamic sign recognition with many words without losing accuracy. In the SSL, practically all the signs are dynamic, so their treatment is crucial when it comes to transcribing gestures into words.

Another aspect that the article addresses is the use of machine learning with respect to the neural networks used by the projects described later. The use of this technique substantially improves the accuracy results with regard to other related works. 

### 1.4. Structure of the Paper

For the development of this article and to achieve the results obtained, a strict re-search process has been followed. In Section 2, the materials and methods necessary for the project have been analyzed, considering the use of a volumetric sensor such as the Leap Motion as a stable and economical option for the execution of the investigation. This section also includes the database of the samples used and the parameters of the volumetric sensor that have been used. The last part of this section describes the use of dynamic time warping (DTW) and how the classifier is implemented.

Section 3 shows the procedures used and describes the different quality parameters for the subsequent analysis of results.

The results are presented in Section 4 where the percentages of the different quality parameters of the research are shown.

In Section 5, the results obtained are compared in a table with those of other published research. Finally, Section 6 presents the conclusions with an accuracy percentage of 95.17% for 176 words, with the number of words implemented being greater than from other publications.

## 2. Materials and Methods

### 2.1. Introduccion

In the first part of this issue, we explain how the Leap Motion Controller works. At the second part, we have introduced a table with the 176 signed words separated into three categories: medical words, verbs, and everyday words. In the final part, the number of samples is commented. The following section explains how the data has been processed according to the position and movement of the hands. Pattern generation explains the methods used for generation. The following section explains how the comparator and data treatment work using the DTW technique. In the last section, the system is implemented using the different Machine Learning Python libraries.

### 2.2. Leap Motion Controller

This commercial device can track the movement of the forearms, hands and fingers in real time. It is a compact, versatile, and economical device that contains 2 cameras with an angle of 120° and three infrared LEDs. It can work at 200 fps (frames per second) and adapts to the amount of light available at all times. This ensures that the device maintains a constant image resolution. It has a USB 3.0 connection and is compatible with Windows, Linux, and Mac. The company that develops it provides the necessary APIs to program in Python, Java, C++, C#, Objective-C and JavaScript. To access the Leap Motion service, C is used.

The Leap Motion controller locates the position of the hands and fingers and transfers the frames to the equipment by means of the USB. On transfer, it returns a type of Hand object (see Figure 1). Each type of Hand object has different subclasses, Arm and Fingers. At the same time, each finger has subclasses of each of the phalanges of the fingers. The different parameters they provide are those related to direction, position, orientation, length, width, and speed.

### 2.3. Acquisition

The implemented system has several parts. First, data is collected by using the leap motion sensor on the computer (see Figure 2).

According to Figure 2, 176 dynamic signs have been recorded. The Spanish name of each sign (word) can be observed in Table 1. Each word has been recorded in 4 different sessions separated by at least 10 days between each session. This has been done to gain independence of the recorded data. The first three sessions were carried out by the same person whilst the fourth session was performed by 15 anonymous people, who are totally independent to the participant of first three sessions.

The implemented acquisition system uses Leap Motion as a hardware device and the tracking software offered by the manufacturer to extract the different parameters. In order to collect information from the hands, the Leap Motion is placed in front of the hands on a support designed for this purpose with a 3D printer. The connection of the Leap Motion with the equipment is made through a USB 3.0 BUS.

The manufacturer-supplied software is a service available on Microsoft Windows or a daemon on MAC and Linux. The current versions of this software offer an API developed in C language, LeapC, which allows access to the data of the service.

Next, a software was developed that works as a user interface so that deaf people could sign and, at the same time, record each of the samples necessary to create the dictionary of signs. In order to make the recordings as accurate as possible, a series of steps were implemented within this proposal to make recording easy for them and to avoid unwanted movement.

A text file is obtained for each word (dynamic sign). Each one of them is defined by 276 parameters or values. These values correspond to the movement of the hands and their position at any given time. At the same time, for the validation of each sign, another software capable of visualizing the movement of the hands and verifying their movement was developed.

At the end of this entire process, a total of 5780 samples were obtained. Each sample corresponds to a plain text file where the data of all the parameters of the LMC are recorded. In each file, there is a different number of frames according to the duration of each sign. In this case, the number of frames also depends on the sampling frequency. This frequency cannot be controlled to a fixed value because it will depend on hardware factors such as the type of port, microprocessor, etc., and on software factors such as the operating system. For example, the hello sign (see Figure 3) is sampled at 110 Frames per Second, fps, for each hand.

Finally, the dataset is composed by 5780 samples, of which 3520 samples have been applied for training and the rest, 2260 samples, for checking the outputs and defining the accuracy.

Spanish sign language omits all prepositions, and a sentence consists of a verb (always in infinitive) and some nouns and adjectives. In this project, authors worked on the recognition of those signs, individually. In the future, a module for processing natural language will be added. The first step is to know whether this proposal can recognize those dynamic words (signs) better than the state of the art and with a larger set of dynamic signs.

The acquisition of data has been applied under the Declaration of Helsinki as a statement of ethical principles for the research involving human subjects on identifiable human data since the sign language is performed by humans. The data is acquired under the consent of the user by anonymous caption. Moreover, the sensor only acquires the movement of the hands by 3D series temporal, and with that information it is very difficult to identify the user. The state of the art does not show references about the human identification by 3D series temporal of the hand movement.

### 2.4. Data Preprocessing

The proposal has been developed using the Python programming language. This language provides a large set of libraries in the field of AI.

First, the signs corresponding to a file containing multivariable time series are loaded into the memory, that is, each sign has an associated matrix where the *y*-axis is the number of frames, and the *x*-axis are the variables. Each variable corresponds to a field in the Leap Motion. Each sign has a variable number of frames (see Figure 3), since each one has a different duration; obviously, they all have the same number of variables.

The parameters come from the Frame object which is, in turn, the root of the data model and provides access to all tracked entities. This way, a frame is created at each update interval. This frame contains attached lists of the hands and fingers corresponding to the instant in which it was created. In addition, the device allows the acquisition of the fingers of a specific hand from the corresponding Hand object. The basic features common to fingers are defined in the Finger class. On the other hand, the Arm object describes the position, direction, and orientation of the arm to which a hand is attached. Finally, the Bone object represents the position and orientation of a bone. The bones tracked include the metacarpals and the phalanges of the fingers (see Figure 4).

In addition to the tracking model features, Leap Motion incorporates Image objects, which provide the raw sensor data and calibration grid for the device’s cameras. 

All the elements presented in Table A1 are derived from the model described above. As can be seen, regardless of elements that provide time information (timestamp, visible time), aspects such as speed, orientation, direction, normal and width of the palm are also considered (see Figure 5); identifiers of frame (id), hand and finger; distance and angles between fingers, and finally, widths, rotations, and joints of all the bones.

In cases of occlusions with the hands, a predictive method is performed, which is quantified in terms of reliability in the confidence parameter. The coordinate axes are established according to Figure 6. Logically, their directions will be defined according to the placement of the device.

### 2.5. Pattern Generation

#### 2.5.1. Parameters

If all the parameters are used, the system can be slowed down in excess. Next, a study was carried out where it was verified that not all the fields are necessary. Finally, it was found that the detection was 100% reliable.

The sensor generates the information of the movement of the hands in a text type file which contains 276 parameters (see Table A1). In order to make the system faster, the number of parameters was reduced. Many of these parameters are not necessary, such as the first one that indicates the frame number. After analyzing all the parameters and carrying out different tests to select the most important parameters, 74 were selected (see Table 2). These parameters are the ones considered fundamental. The number corresponds directly to Table A1 in the Appendix A.

As can be seen, the parameters corresponding to the rotations in the x, y and z axes of all the fingers have been chosen. To reduce the complexity of the system, the hands were separated during the generation of the patterns.

The selection of parameters is based on keeping the rotation, translation, and size invariance. Thus, the proposal can be applied in different places, with a different positioning of the sensor and environment conditions. Therefore, the user can use the device in any situation. This will be important for the recording of the different sessions of the dataset.

#### 2.5.2. Generation

The independence of the samples has been established by making the recordings as follows: three sessions of 10 samples each were recorded, plus an additional session with samples recorded by people different to the participant of the first three sessions. These sessions were held at different times, that is, with a temporary space of at least 10 days. Once an analysis of the time of the project was carried out, it was considered that 176 words could be recorded. To analyze the evolution of the process, the results were verified in sets of 50 words. Checks were made at 50, 100 and, finally, at 176 words. Finally, the data of the fourth session, consisting of 50 words and 10 samples per word, have been used to validate all processes, with 15 anonymous users who did not participate in the first three sessions.

Initially, the sampling frequency is variable depending on the hardware and software conditions of the equipment. Before generating the patterns, it is necessary to equalize the size of the files, so that they have the same number of frames in each of them. Each file corresponds to a sample of each word.

The person who recorded each of the words was a specialist in Spanish sign language. To create the patterns, 20 samples from 2 different sessions have been used. One word Q discomposed by 20 observations from *Q1* to *Q20*.

For the generation of the pattern, the shortest series was searched first, causing the rest of the recordings to be compressed to the size of that series. They were then added together and divided by the number of existing samples creating the average pattern.
(1)$$\frac{\sum_{i=1}^{i=20}Q_{i}\left(x\right)}{20}$$

In the end, the pattern keeps the form of the signed word and also adds features of each of the references, adding robustness to the system, so that it continues to be reliable for different sessions of the same user, as well as between different users.

It is very important that when generating the patterns, it is taken into account that the recordings are made correctly, verifying that they are correctly recorded and that the files have a similar size.

### 2.6. Comparator

#### 2.6.1. Introduction to DTW

Dynamic time warping (DTW) is a classification technique that is used in many fields of research. It is widely used to compare time series of different lengths. A field where it is widely used is in voice recognition where each person, when speaking, can say the same thing, yet not everyone speaks at the same speed. With DTW this time warping is resolved.

There are instances where DTW is used. For example, Tuzcu [35] uses it for the automatic recognition of ECG signals; Legrand [36], for chromosome recognition; Kovacs-Vajna [37], in fingerprint recognition; whilst Rath [38] uses it in the development of the recognition of handwritten documents. Research using DTW for signature recognition can also be found in [39,40]. Within the investigations based on voice recognition are [41,42]. A field in which it is also used is in the recognition of facial [43] and body gestures [44]. In the recognition of gestures there are many investigations where some of them use the Kinect [45,46,47,48]. Finally, within the research that deals with the recognition of sign language are [49,50,51].

#### 2.6.2. Development of the DTW 

When a person is signing, the speed with which they move their hands changes not only with other people, but with themselves, in the same way that sometimes a person speaks faster or slower. In order to compare time series of different lengths, the DTW technique was used. If there are 2 series in time K and T, with their lengths M and N and considering that each frame captured by the sensor, as an element of the function, what remains for all the captured frames is:K = k_1_, k_2_,……, k_i_,……, k_n_(2)
T = t_1_, t_2_,……, t_i_,……, t_n_
(3)

Assuming that k = t, it is not necessary to calculate the distance of the 2 sequences. If k ≠ t the alignment is further apart and there is no other choice but to try to align both sequences. To align both sequences it is necessary to build an M × N matrix, whereby each element of a matrix (I, J) represents the point Ki and Tj, the alignment.

Now the path is defined as a deformation path of the regular path, and W is used (see Figure 7) to indicate that the KK element of W with the final result of Wk = (i,j)k, at the end the following is obtained:W = w1, w2,…,wk,….,wk  max(m,n) ≤ K < m + n − 1(4)

At the same time, the following conditions must be met:
Boundary condition; w1 = (1, 1) y wK = (m, n)Continuity; Si wk-1 = (A’, B’), then the next point W for the path k = (A, B) must comply (A-A’) <= 1 y (B’) <= 1Monotone; Si wk-1 = (A’, B’), then the next point W for the path k = (A, B) must comply 0 <= (A-A’) y 0 <= (B-B’).

The final objective is to extend and shorten the two time sequences and establish the shortest distance, so that in the end the distance γ(i,j) is minimal.
γ(i,j) = d(qi,cj) + min{ γ(i − 1,j − 1), γ(i − 1,j), γ(i,j − 1)}(5)

In the graphic example of Figure 8a, a path generated by the DTW is shown, corresponding to a word. Figure 8b represents a sign of the same class, and it can be verified that both are similar. On the other hand, in Figure 8c, it can be seen how a different sign of the pattern behaves.

#### 2.6.3. Comparator Scheme

The algorithm used to compare the different signs follows the structure of Figure 9. Each one of the 10 samples of each word is compared with the patterns. The output will show the result for the shortest distance from the input.

The *“Reference Model”* are each of the patterns generated above, using 20 samples from 2 independent sessions. Each session has a set of 10 samples.

Since the sampling frequency is high, before entering the data in the comparator so that the system does not slow down and at the same time eliminate noise, a decimation is performed on the samples and patterns.

### 2.7. Implementation

For the implementation, a set of techniques have been chosen using the DTW as a comparison metric. DTW is an algorithm that allows the comparison of time series of different lengths. This algorithm is straight forward to implement, and much information is open for use by researchers. Specifically, in Python there is a package called tslearn that already has this algorithm implemented. This package has already been tested and proven by many users. Finally, note that the implementation has been carried out using machine learning. The decision to use this type of technology was made because other investigations already had neural networks implemented and their results could be improved with this technique.

The Python packages used in the investigation are numpy, pandas, os, tslearn, and sklearn. Numpy and pandas are used for data loading and processing. The package helps manage files. This module is very important since, as mentioned in Section 2, each of the samples corresponds to a file. The tslearn package provides automatic processing tools for time series analysis. Specifically, the tslearn.metrics.dtw library has been used. To generate the confusion matrix and obtain the quality parameters of the system, accuracy, F1 Score, recall and precision, the sklearn package has been used.

## 3. Experimental Methodologies

### 3.1. Procedures

At first, different greetings were recorded: good morning, good afternoon, good evening, and hello. From the beginning, the aim was to reach a minimum of words. Due to time and progress, the final goal was 176 words with different meaning, whereby most of those words aimed at establishing a medical conversation between a deaf person and a doctor. To check the evolution, 50-word tests were carried out. The tests were carried out at 50, 100 and, finally, at 176 words. Finally, the samples (signs) from the fourth session will be applied for testing purposes only, and hence, to validate the proposal and demonstrate that the models generated by one person can be used by different people whilst the accuracy of the proposal is similar. 

### 3.2. System Quality

In order to quantify the experiments performed with the 3-word sessions, quality measures need to be used. To obtain the results, the Python Scikit-Learn libraries were used.

In this way, 10 samples were obtained for each of the 3 sessions. Two sessions are used to generate the patterns and 1 session to compare it within the system. This ensures session invariance. Finally, at the output of the comparator, the confusion matrix and the quality parameters that provide us with the efficiency of the system are obtained (see Figure 10).

In addition, the proposal performed a fourth session by anonymous users different to the user from the previous first three sessions. The session is composed by 50 words and 10 repetitions per word. Hence, the proposal can validate the models generated by each sign and their use for different people.

#### 3.2.1. Confusion Matrix

From the confusion matrices, the different parameters necessary to measure the reliability and quality of the system are calculated. Figure A1 shows how the confusion matrix behaves. The diagonal shows the success of each of the words by itself. The green boxes correspond to several hits from 8 to 10, both inclusive, whilst the orange boxes show a hit rate of 5 to 7, both inclusive. Meanwhile, the boxes marked in red show a success rate of less than 4. Off-diagonal, it can be verified how there are different crosswords where the system does not recognize the pattern of the same word but does recognize the pattern of another word.

In Figure 11, Reality corresponds to the input and Prediction corresponds to the output. The different parameters are described below:False Positive [FP]; A False Positive occurs when the model expects an output, and it does not occur.False Negative [FN]; A False Negative occurs when the model does not expect an output that is eventually produced.True Positive [TP]; A True Positive is when the model expects an output, and it does.True Negative [TN]; A True Negative is when the model does not expect an output and it does not eventually produce one.

#### 3.2.2. Precision

Using the accuracy metric, the quality of the model can be measured. This returns the percentage of hits that are expected.
(6)$$\mathrm{precision}=\frac{\mathrm{TP}}{\mathrm{TP}+\mathrm{FP}}\times 100$$

#### 3.2.3. Recall

This measure informs about the quantity that the model is able to identify.
(7)$$\mathrm{recall}=\frac{\mathrm{TP}}{\mathrm{TP}+\mathrm{FN}}\times 100$$

#### 3.2.4. F1

This feature combines the precision and recall parameters. This parameter helps compare the combined performance of both parameters. It is calculated by performing the harmonic mean of precision and recall.
(8)$$F1=2\cdot \frac{\mathrm{precision}\cdot \mathrm{recall}}{\mathrm{precision}+\mathrm{recall}}\times 100$$

#### 3.2.5. Accuracy

This parameter measures the system’s percentage of success and its formula would be:(9)$$\mathrm{accuracy}=\frac{\mathrm{TP}+\mathrm{TN}}{\mathrm{TP}+\mathrm{TN}+\mathrm{FP}+\mathrm{FN}}\times 100$$

## 4. Results

### 4.1. Experiment

There are two fundamental aspects to carry out the experiment: to see the success of the set of words, and to see the evolution of success as the number of words increases.

In the initial phase, the recordings were made, and the success rate and the possible word crossings were checked. For the first 50 words, an accuracy of 98.80% was obtained. The quality parameters obtained from the matrices (see Figure A1, Figure A2 and Figure A3) and the parameters are shown in Table 3. With 100 words, the accuracy was 97.60%, where its confusion matrix can be verified in Figure A2. The system dropped approximately 1 point with the addition of 50 words. Finally, it was tested with the total set of the database of 176 words. The accuracy of the implemented system dropped to 95.17%, with its confusion matrix in Figure A3.

### 4.2. Experiment for Validation

In order to show the robustness of the proposal and validate the use of this approach, a fourth session of data was built. Fifty words with 10 repetitions per word were recorded by 15 anonymous users, totally different to the user of the three first sessions. The models built for two of the three first sessions were used for this fourth session of samples. The results are shown in Table 4. The accuracy of the experiment was up to 94.80%.

## 5. Discussion

The results show that the efficiency of the system is above 95% in all its phases. As words are added, the system’s efficiency decreases, which may become an issue in recognizing some words. The approach has been validated by a session carried out by different users than that for building the models of each sign or word. The quality metric decreased 4% for accuracy, 3.59% for precision, 4% for recall and 4.03% for F1 between the user that built the models of signs and the new 15 anonymous users, who have validated those models. These results show that accuracy slightly decreases (between 3.59% and 4.03%, according to the quality metric) for different users or signers, and the models are independent of the person doing them. Therefore, this approach has inter-user invariance.

On the other hand, Table 5 compares the results obtained in this paper with the results of other publications. It should be noted that regardless of the number of words used in this paper, there are several differences with the research described in Table 5. These proposals are aimed at the three trends indicated: use of static signs, use of static and dynamic signs, and use of dynamic signs. The limitation in the number of signs is observed. This research proposes a system based on a pattern generator and the DTW to demonstrate that the number of words can be increased while maintaining success rates.

It should be remembered that 20 samples from two independent sessions were used to create the patterns. If someone wants to improve efficiency so that the pattern picks up hand movements better, the number of samples to generate a pattern should be increased. With 50 samples recorded in different sessions and with different people, the pattern would be more robust, and the efficiency of the system would improve significantly.

It should be noted that the proposal improves against different Deep Learning techniques and, in general, machine learning, which can obtain good results, but for a limited number of words, between 10 and 50 signs and/or words. This proposal includes 176 dynamic words, thus demonstrating that in comparison to dynamic signs, this proposal maintains a positive robustness.

In future research, the aim will be to increase the number of words with problems related to similar signs that require another type of treatment, for instance, the case of the words disappearing and breaking. Both signs have a similar behavior regarding the movement of the hands, meaning that context is sometimes needed to be able to identify them separately.

In Spanish sign language, there are words that are signed exactly the same and that only change their meaning due to the shape of the face. In those instances, it would be useful to include a camera that would interpret the shape of the face if necessary.

Finally, note that in order for a system like this to be useful in real life, it would require approximately 1000 words of common use. For this, efficiency would have to be improved, as mentioned above.

## 6. Conclusions

This paper shows a word recognition system for the Spanish sign language. A pat-tern creator is applied, and DTW is used to establish the correct word, achieving a success rate of 95.17% for a total of 176 dynamic signs, improving the number of words of the state of the art, which were between 10 and 50 dynamic signs. At the same time, the recognition of dynamic signs has been improved, applying automatic recognition techniques using the DTW. In addition, it improves systems using deep learning and machine learning techniques, in general, showing this proposal to be more robust than the techniques used until now.

This proposal has validated the use of sign models developed by a user and then used by different users or signers. Therefore, the proposal shows an invariance inter-user. This is an added value of the proposal because it keeps the accuracy of 94.8% with inter-user validation vs. the 98.80% for the inter-session validation with the same user with 50 words. From a practical point of view, the model created by one user or signer can be used by any other user or signer.

## Figures and Tables

**Figure 1 sensors-23-00481-f001:**
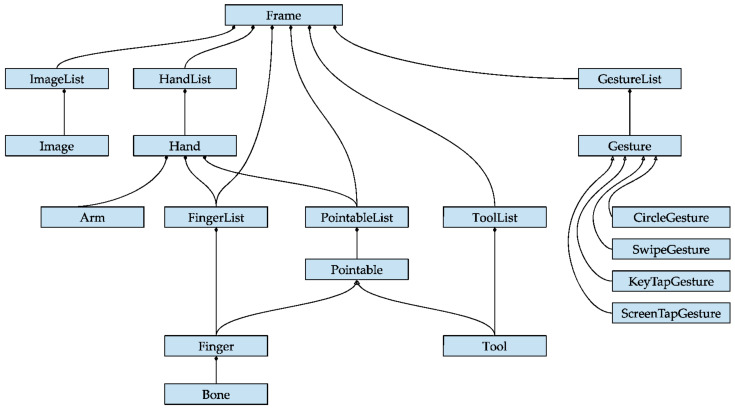
Frame Object of Leap Motion. Source available online: https://www.researchgate.net/figure/Frame-Object-of-Leap-Motion-Source_fig5_342433909 (accessed on 29 December 2022).

**Figure 2 sensors-23-00481-f002:**
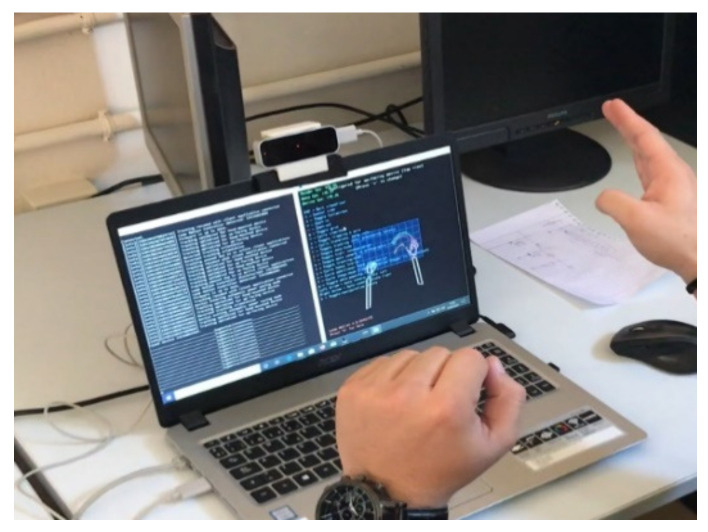
Recording of samples.

**Figure 3 sensors-23-00481-f003:**
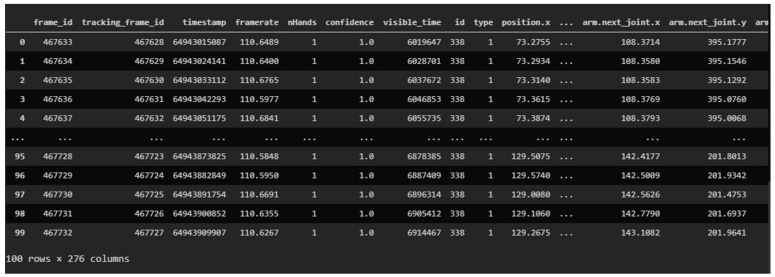
Hello sign data.

**Figure 4 sensors-23-00481-f004:**
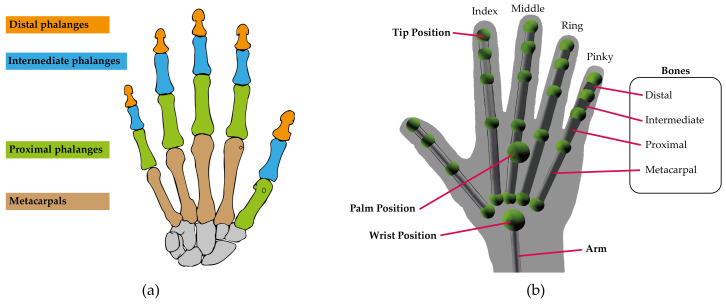
Tracking of the hands skeletal model (**a**) Object Bone. Source available online: https://developer-archive.leapmotion.com/documentation/javascript/devguide/Leap_Overview.html (accessed on 29 December 2022) (**b**) Object Hand. Source available online: https://ieeexplore.ieee.org/abstract/document/8538425/figures#figures (accessed on 29 December 2022).

**Figure 5 sensors-23-00481-f005:**
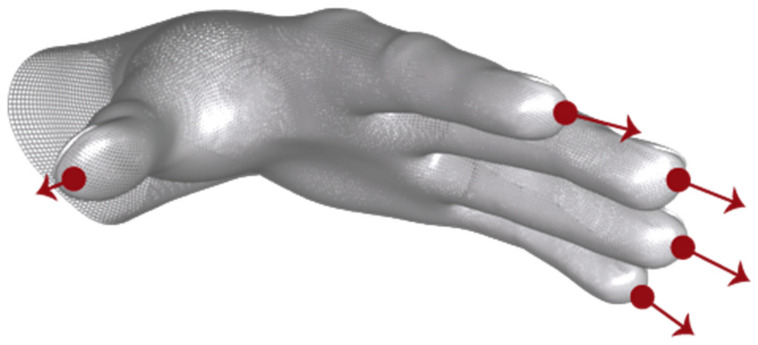
Palm Movement. Source available online: https://developer-archive.leapmotion.com/documentation/javascript/devguide/Leap_Overview.html (accessed on 29 December 2022).

**Figure 6 sensors-23-00481-f006:**
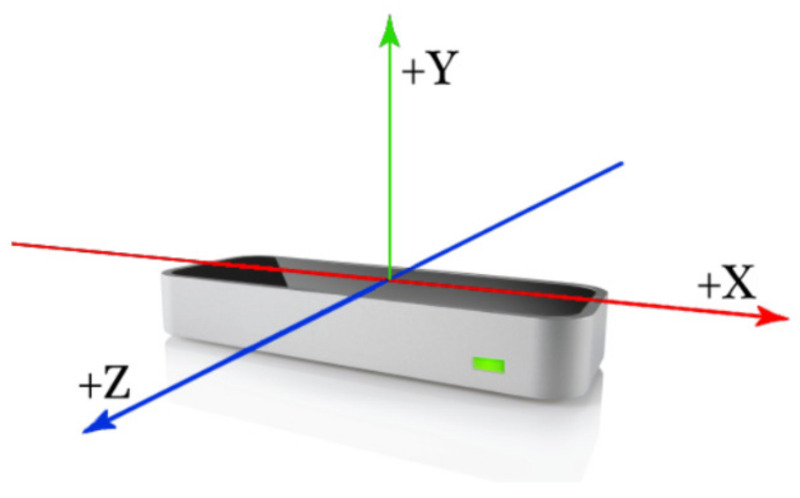
Leap Motion axes. Source available online: https://developer-archive.leapmotion.com/documentation/javascript/devguide/Leap_Overview.html (accessed on 29 December 2022).

**Figure 7 sensors-23-00481-f007:**
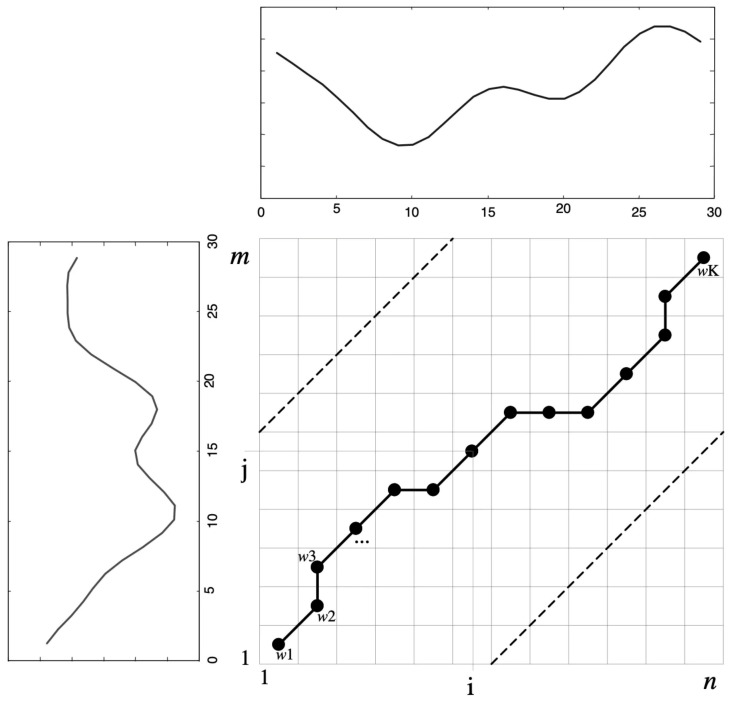
Example of a warping path. Source available online: https://blog.csdn.net/niyanghuahao/article/details/78612157?locationNum=9&fps=1. (accessed on 29 December 2022).

**Figure 8 sensors-23-00481-f008:**
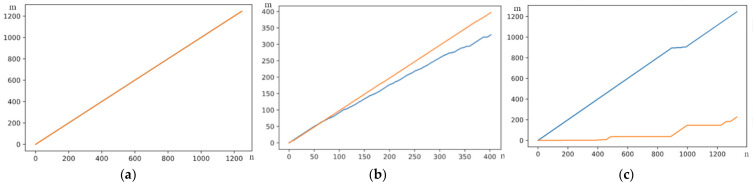
Comparator DTW. (**a**) Word Path of Pattern. (**b**) Comparison of a word (blue) sample with its own pattern (orange). (**c**) Comparison of a word (orange) other than the pattern (blue).

**Figure 9 sensors-23-00481-f009:**
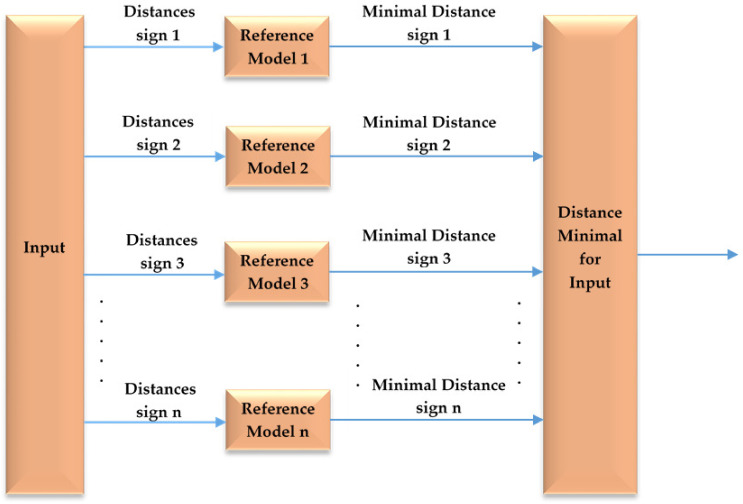
Comparator scheme.

**Figure 10 sensors-23-00481-f010:**
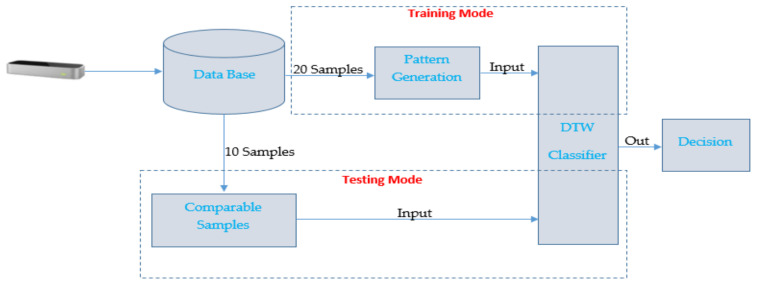
Implemented system.

**Figure 11 sensors-23-00481-f011:**
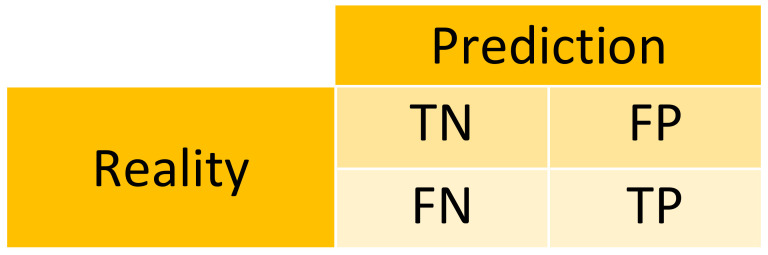
Graphic model of the parameters.

**Table 1 sensors-23-00481-t001:** Set of recording words.

Medical Words						
Allergy	Alzheimer	Ambulance	Anxiety	Asthma	Bacteria	Bladder
Blood	blood circulation	Blood Test	Breasts	Burp	Cancer	Care
Chest	Consult	Depression	Diabetes	Doctor	Fainting	Fatten
Fever	Gluteus	Headset	Health	Heart	Heart Attack	Hemorrhage
Hormone	Hospital	Ictus	Implant	Inflammation	Injection	Injury
Intestine	Jaw	Liver	Lumbago	Lungs	Mask	Medicines
Organs	Ovaries	Oxygen	Phlegm	Pressure	Prostate	Serum
Sickness	Stitches	Stomach Stress	Stress	Swelling	Urgency	Vaccine
Vagina						
Verbs						
I didn’t know	I don’t know	Is	To Ask	To Break	To Burn	To Cook
To Cure	To Disappear	To Do	To Eat	To Fall	To Feel	To Go
To Have	To Operate	To Reduce	To Rest	To Run	To Shower	To Size
To Take	To Take a walk	To Try	To Walk	To Work		
Everyday Words						
Accident	After	Air	All Right	Already	Also	Always
April	August	Back	Before	Car	Centre	Coat
Cold	Danger	Day	Deaf	December	Ear	Effort
Elbow	End	Example	Eyelid	Eyes	Family	Fat
February	Feet	Friday	Gloves	Good	Good Afternoon	Good Bye
Good Morning	Good Night	Hands	Head	High	Higher	Hour
How are you?	Hungry	Information	June	Last Night	Little	Lonely
Man	March	Midday	Moto	Mouth	Much	Mummy
Neck	Night	No	No Problem	Noise	Nothing	Now
October	Panic	Private	Rain	Regular	Risk	Saturday
Sensation	September	Sex	Shape	Shoulder	Something	Suffer
Sure	Thank You	They	Thin	To	Too	Trouble
Unusual	Upset	Wednesday	Without	Yesterday	Yours	

**Table 2 sensors-23-00481-t002:** Number of parameters used for the system according to Table A1.

Parameters								
4	8	9	10	11	27	28	29	32
41	42	43	52	53	54	63	64	65
74	75	76	79	88	89	90	99	100
101	110	111	112	121	122	123	126	135
136	137	146	147	148	157	158	159	168
169	170	173	182	183	184	193	194	195
204	205	206	215	216	217	220	229	230
231	240	241	242	251	252	253	262	263
264	272							

**Table 3 sensors-23-00481-t003:** Quality parameters.

Nº Words	Precision	Recall	F1
50	98.93%	98.80%	98.80%
100	97.96%	97.60%	97.59%
176	96.32%	95.17%	95.02%

**Table 4 sensors-23-00481-t004:** Validation of sign models.

Nº Words	Precision	Recall	F1
50	95.34%	94.80%	94.77%

**Table 5 sensors-23-00481-t005:** Set of results of papers that using LMC.

Reference	Gestures	Data Set	Classifier	Accuracy
Static Signs				
[18]	24 ASL Letters	unclear	Decision tree	82.71%
[19] *	10 ASL	14 people × 10 sets × 10 samples	SVM	80.86%
[20]	24 GSL	6 people × 10 sets	MLP (boneTraslation)	99.08%
			MLP (palmTraslation)	98.96%
[21]	32 ASL	146 people × 1 sample/letter	MLP	90%
[22]	24 ASL Letters Use sentences	12 people × 10 samples	Linear Regression Analysis	86.1%
Static and Dynamic Signs				
[23]	28 ArSL	10 samples/letter	KNN	97.1%
			HMM	97.7%
[24]	28 ArSL	10 samples/letter	Naïve Bayes	98.3%
			MLP	99.1%
[25] *	16 ArSL Static Words 20 Dynamic Words	2 people × 200 samples	Static	
			Neural Network	90.35%
			K-NN	95.22%
			SVM (RBF Kernel)	89.12%
			SVM (Poly Kernel)	90.78%
			Dynamic	
			DTW	96.41%
[26]	26 ASL letters	4 people × 20 samples/letter	MLP-BP	96.15%
[27]	26 ASL letters 10 digits	12 people Unclear samples	SVM (letters)	80.30%
			DNN (letters)	93.81%
			SVM (total)	72.79%
			DNN (total)	88.79%
[28]	26 ASL letters	100 people × 1 samples/letter	LSTM	97.96%
			SVM	98.35%
			RNN	98.19%
[29]	26 ASL letters	5 sets × 450 samples/letter	CNN	80.1–99.7%
[30]	26 SIBI letters	5 people × 10 samples/letter	K-NN	95.15%
			SVM	93.85%
[31]	15 ArSL words	5 people × 10 samples/letter	LSTM	96%
[32]	18 static 12 dynamic	20 people × 60 samples/word	LSTM	96.41%
Dynamic Signs				
[33]	50 ArSL	4 people × 1 set	MLP	88%
[34]	13 ASL words	10 samples	Neural Network	99.9%
			Random Forest	99.7%
			SVM	99.9%
			K-NN	98.7%
			Naïve Bayes	96.4%
This Paper	50 SSL words	1 people × 30 samples/word	DTW	98.80%
This Paper	100 SSL words	1 people × 30 samples/word	DTW	97.60%
This Paper	176 SSL words	1 people × 30 samples/word	DTW	95.17%
This Paper	50 SSL words	15 different people × 10 samples/word	DTW	94.80%

* Papers that used Kinect and LMC.

## Data Availability

The data presented in this study are available on request from the corresponding author. The data are not publicly available due to they is being analyzed for future publications before to openly available.

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
