# Peer review of "Robust Identification System for Spanish Sign Language Based on Three-Dimensional Frame Information"

_sensors, 2023, doi:10.3390/s23010481_

Round 1
Reviewer 1 Report (New Reviewer)
Dear authors,
Your paper about the description of a technique to transcribe the Spanish Sign Language is quite original and easy to read. The scientific soundness is high as well as the interest to the readers. However, there a few aspects to improve in order to be published:
- Chapter 2, materials and methods, starts mentioning the commercial device. It is recommended to introduce better this chapter and explain what you are going to do.
- English Grammar. There are a few paragraphs difficult to understand, as for example: "according to World Health Organization (WHO) of the world´s people has some kind of hearing problem". It is recommended to review the whole paper, there are many examples which requires rewrite.
- There are some format issues. For example, chapter 2.4.1 title is different than 2.5.1. There are several cases like this.
Kind regards.
Author Response
Please check the attachment, thank you.

Reviewer 2 Report (New Reviewer)
I reviewed the manuscript tittle Robust identification system for Spanish sign language based 2
on three-dimensional frame information
The paper is interesting and discusses an import topic for many parties and beneficiaries.
High quality paper with rich content
Many readers practitioners can benefit from this paper. However, the authors need to consider the following changes and comments to make the paper ready for publication some
Introduction
Authors need to provide some support/references for the first paragraph and where the data have been sourced, although UN was mentioned
Authors need to check the citation and make sure all references have been cited correctly
The introduction is so long but I believe it has a lot of important information. Many parts are related work and literature review. Therefore, it has to be restructured/organized with some subheadings to make it easy for reader understand such important parts, authors might need to restructure the introduction part (page 1-page 4)
In the paper
There are a lot of tables and figures either do not make significant contributions or can be merged, see for example the following
Table 1. Set of recording words
This table can be reorganized, for example the columns must have some labels or the words can be classified to better understand them
The paper has two figures with same number unless authors meant something else
Figure 8. Example of a warping path 384
Figure 8. Word Path of Pattern. 402
Anyway, some figures can be removed or merged in page 12 and 13 such as
Fig 8, fig 9, 10 ….etc
The same is true for page 15-16 figures 13-16 do not make sense or need to be merged
Again tables Table 3. Quality parameters and Table 4. Validation of sign models
Can easily be merged
Table 5. Set of results of papers that using LMC
This part is very confusing and too long table makes it hard to follow and understand
Conclusions
This part is poor and authors were very brief where they could have more drawn conclusions to enhance the value of their contributions and show the value of the work. I believe the work has a lot of other significant and brilliant conclusions that can be presented here
Author Response
Please check the attachment, thank you.

This manuscript is a resubmission of an earlier submission. The following is a list of the peer review reports and author responses from that submission.
Round 1
Reviewer 1 Report
Review of the article "Robust identification system for Spanish sign language based on three-dimensional frame information".
In my opinion this is an interesting article, but the Authors should better emphasize their contribution. The heart of the solution is Leap Motion Controller, which provides information about three-dimensional frame. This information is processed by the software implemented by the Authors.
There is information in the article (page 6): "All the software in this section has been developed in the Python programming language", but it is not stated what functionalities have been implemented in Python. It is known that the deep learning techniques and machine learning have been used, but the entire implementation related to artificial intelligence techniques is treated as "a black box".
More information about deep learning techniques and machine learning should be given - as I suppose, an artificial neural network has been implemented (how many layers there are in this network, how many neurons, how the network has been taught?).
There is also information in the article (page 6): "This language provides a large set of libraries in the field of AI" - information should be provided in the article about what other artificial intelligence techniques have been used in the Authors' software. Information should also be given as to whether ready-made Python modules were used, or whether the entire software was implemented on their own by the Authors.
The implemented software can recognize single words. Can the software recognize these words when they are in a sentence (then it may be a very complicated task to determine when a given word begins and when it ends)?
Author Response
"Please see the attachment."

Reviewer 2 Report
1. Does the article focus on the Sensors aim and scope?
2. The paper structure should be described at the end of the introduction section.
3. The introduction is too weak. The introduction should indicate the research gaps and research goals. What is the main question addressed by the research?
4. What does it add to the subject area compared with other published material?
5. The authors are suggested to cite related works to illustrate your equations. And should make a connection with these equations. Why do you use related theorem?
6. What specific improvements could the authors consider regarding the methodology?
7. In the Methodology section, the authors should use a flowchart to describe the proceeding flowchart. Please use a standard flowchart to illustrate the process. For example, use an elliptical graph to illustrate the “start” and “end”, use the diamond graph to illustrate the judgment events. Figure 12 is not a standard flowchart.
8. The implementation environment is not clear.
9. Your contributions should be presented in conclusion.
10.In fact, the authors only list the specific keywords in title. Too long title is not proper.
11. I still think the introduction should indicate the research gaps and research goals and the main question addressed by the research is enough.
12. A paper just simple solve 1-2 past unsolved research gaps is enough.
Author Response
"Please see the attachment."

Reviewer 3 Report
This paper presents a research on DTW on the 3D hand movement data captured by the Leap Motion volumetric sensor for Spanish Sign Language Recognition. The method is validated on a dataset and achieves an accuracy of 95.17. I have the following detailed comments:
1. The novelty of this paper is very limited. There are many existing works on sign language recognition. The proposed method utilizes an existing technique, DTW, to solve a popular problem.
2. The authors only conduct experiments on a very small dataset, where one person only conducts the sign language.
3. The writing of this paper can be greatly improved. It is more like a technical report rather than a quality research paper.
Round 2
Reviewer 1 Report
The article has been corrected properly and carefully, taking into account all my concerns and comments. Now, in my opinion, the article can be published in Sensors.
Reviewer 2 Report
The authors have fixed previous concerns.
Reviewer 3 Report
The authors made some efforts in improving the paper, but I still find that the paper is below the standard of a quality research to be published in Sensors.
1. The novelty of this paper is still very limited. None of the contributions claimed by the authors can be seen as a solid technical contribution. The core technique, DTW, is a well-developed technique, and a bit old as well.
2. The dataset is very small. There is only one person performing all the gestures. So it is difficult to check whether the method could generalize to multiple people. Usually, at least 40 or more people should be involved in the dataset.
3. Usually I will not raise new questions for revised paper, but Section 2.5.1 is empty. Please proofread the paper careful before submission.